# Laboratory correlates of SARS-CoV-2 seropositivity in a nationwide sample of patients on dialysis in the U.S.

Shuchi Anand[1]*, Maria E. Montez-Rath[1], Jialin Han[1], Pablo Garcia[1], Julie Bozeman[2], Russell Kerschmann[2], Paul Beyer[2], Julie Parsonnet[3,4], Glenn M. Chertow[1,4]

1 Division of Nephrology, Department of Medicine, Stanford University School of Medicine, Palo Alto, California, United States of America, 2 Ascend Clinical Laboratory, Redwood City, California, United States of America, 3 Division of Infectious Diseases and Geographic Medicine, Department of Medicine, Stanford University School of Medicine, Palo Alto, California, United States of America, 4 Department of Epidemiology and Population Health, Stanford University School of Medicine, Palo Alto, California, United States of America

* sanand2@stanford.edu

**Data Availability Statement:** The data underlying this study are available on Dryad (https://doi.org/10.5061/dryad.wpzgmsbmr) and from Stanford

## Abstract

Patients on dialysis are at high risk for death due to COVID-19, yet a significant proportion do survive as evidenced by presence of SARS-CoV-2 antibodies in 8% of patients in the U. S. in July 2020. It is unclear whether patients with seropositivity represent the subgroup with robust health status, who would be more likely to mount a durable antibody response. Using data from a July 2020 sample of 28,503 patients receiving dialysis, we evaluated the cross-sectional association of SARS-CoV-2 seropositivity with laboratory surrogates of patient health. In separate logistic regression models, we assessed the association of SARS-CoV-2 seropositivity with seven laboratory-based covariates (albumin, creatinine, hemoglobin, sodium, potassium, phosphate, and parathyroid hormone), across the entire range of the laboratory and in comparison to a referent value. Models accounted for age, sex, region, race and ethnicity, and county-level COVID-19 deaths per 100,000. Odds of seropositivity for albumin 3 and 3.5 g/dL were 2.1 (95% CI 1.9–2.3) and 1.3 (1.2–1.4) respectively, compared with 4 g/dL. Odds of seropositivity for serum creatinine 5 and 8 mg/dL were 1.8 (1.6–2.0) and 1.3 (1.2–1.4) respectively, compared with 12.5 mg/dL. Lower values of hemoglobin, sodium, potassium, phosphate, and parathyroid hormone were associated with higher odds of seropositivity. Laboratory values associated with poorer health status and higher risk for mortality were also associated with higher likelihood of SARS-CoV-2 antibodies in patients receiving dialysis.

## Introduction

Patients receiving dialysis are at high risk for COVID-19 related hospitalizations [1] and if hospitalized, high risk of death. Studies from Europe [2–5], Latin America [6], and the U.S. [7, 8] report that nearly one-third may die if hospitalized, and up to half may die if they require intensive care [8]. In addition to presenting features of illness—e.g., fever or cough,

Medicine (https://covidkidney.stanford.edu/cross-sectional-data).

**Funding:** This study was supported in the form of a career development grant from the National Institutes of Diabetes, and Digestive and Kidney Diseases (Grant No. 5K23DK101826) awarded to SA. Ascend Clinical Laboratories provided support in the form of salaries for authors JK, RK, and PB. The specific roles of these authors are articulated in the 'author contributions' section. Ascend Clinical Laboratories supported the remainder plasma testing for SARS-CoV2 antibodies. The funders had no additional role in study design, data collection and analysis, decision to publish, or preparation of the manuscript. No additional external funding was received for this study.

**Competing interests:** The authors have read the journal's policy and have the following competing interests: JB, RK and PB are employed by Ascend Clinical Laboratories (http://clinical.aclab.com/). GMC is on the Board of Satellite Healthcare, a not-for-profit dialysis organization. There are no patents, products in development or marketed products associated with this research to declare. This does not alter our adherence to PLOS ONE policies on sharing data and materials.

lymphopenia, or elevated serum ferritin—underlying patient health status are risk factors for death [7, 9].

However, hospitalized patients represent a small fraction of patients with SARS-CoV-2 infection, since growing national and international data indicate that only approximately 10% of patients with presence of SARS-CoV-2 antibody are diagnosed as cases [10, 11]. In our recent analysis of remainder sera [12], 8% of patients receiving dialysis had evidence of SARS-CoV-2 S1 spike receptor binding domain (S1RBD) total antibody in July 2020, implying that a sizeable number were infected and survived SARS-CoV-2 exposure. Clarke et al. reported that 40% of patients on dialysis were asymptomatic and/or undiagnosed despite showing evidence for SARS-CoV-2 nucleocapsid antibody [13].

Evaluating correlates of SARS-CoV-2 seropositivity, rather than of diagnosed or hospitalized cases, therefore captures a larger range of affected patients. While poorer health status may be associated with higher risk of death among patients on dialysis hospitalized with COVID-19, it is not known whether poorer health status is also a risk factor for SARS-CoV-2 seropositivity. One major consideration is that SARS-CoV-2 seropositivity implies ability to mount an antibody response to infection, which may be impaired among older and frail patients receiving dialysis [14–16].

Several routinely collected laboratory studies are valid surrogates of health status in patients receiving dialysis [17–20]. In the current study, we investigated the association of routinely collected laboratory tests with the presence of SARS-CoV-2 total antibody. Recognizing the interconnections among overall health, nutritional status, inflammation, and the immune response, we hypothesized that after accounting for community burden of COVID-19, patients with laboratory surrogates of more robust health status (i.e., higher serum concentrations of albumin, creatinine, and hemoglobin) would more likely have survived a SARS-CoV-2 infection and mounted an antibody response than patients with less robust health status. We also explored the odds of seropositivity associated with other routinely measured laboratory tests.

## Materials and methods

In partnership with Ascend Clinical Laboratory, a central laboratory that receives laboratory samples from approximately 65,000 patients receiving dialysis throughout the U.S., we tested a random sample of 28,503 patients for SARS-CoV-2 total antibody in July. The Stanford University Institutional Review Board 61 (Registration 4947) approved this work under study protocol #56901. The data were fully anonymized before Stanford University researchers analyzed them; all sample analyses were performed as part of routine clinical care or using plasma that would have otherwise been discarded (for the SARS-CoV-2 antibody testing). The Stanford University IRB waived requirement for informed consent.

### Sampling procedures, covariates and outcome

Details of our sampling methodology and covariate definitions are published elsewhere [12]. Briefly, in order to attain representativeness of patients on dialysis in the U.S., we used implicit stratification for sampling accounting for age, sex, and U.S. census region (using patient zip code to assign region). The age and sex distribution of patients included in our sample matched the United States Renal Data System reported distribution as of January 1, 2017 [21].

Among sampled patients, we extracted electronic health record data to ascertain demographics, residence zipcode, and comorbidities. Patients' electronic health data on demographics and comorbidities as available to Ascend Clinical Laboratory were accessed for the month of the seroprevalence analysis (July 2020). Patients' laboratory data—performed as part of

routine clinical care—were accessed dating back 6 months prior to seroprevalence testing (January 2020). The source of data were Ascend Clinical Laboratory and electronic medical record data; all data were anonymized prior to analysis.

We assigned each patient to one of four regions using U.S. Census classification of the Northeast, South, Midwest, and West. We also assigned a community burden of COVID19 via linkage to county-level data from the Center for Systems Science and Engineering at Johns Hopkins University [22]; we selected the metric of county cumulative deaths per 100,000 as of June 30, 2020 on the basis of our prior analyses [12]. We used the Siemens Healthineers S1RBD total immunoglobulin assay to define seropositivity [23].

## Laboratory studies

We extracted available data from July for the following routinely collected laboratory studies: serum albumin, creatinine, hemoglobin, sodium, potassium, phosphate, and parathyroid hormone (PTH), all measured at the central laboratory using standardized assays. Albumin was measured using the Bromcresol Green (BCG) method. Ninety five percent of sampled patients had analyzed laboratories drawn on the same date as the SARS-CoV-2 antibody testing. If more than one value were available for a patient during the month, we used the value nearest in date to his or her SARS-CoV-2 antibody testing. In order to evaluate the possibility of reverse causation, that is, if an association with poorer health status and seropositivity emerged, it was due to antecedent infection with SARS-CoV-2 leading to a decline in health status, we selected serum albumin as the laboratory surrogate most likely to be affected by infection and the inflammatory response. Furthermore unlike other potential markers of health status (e.g., phosphate or parathyroid hormone) [24], serum albumin has a strong and consistently observed inverse association with mortality among patients on dialysis [17, 25, 26]. We thus extracted data for serum albumin starting prior to the emergence of SARS-CoV-2 (January 2020).

## Statistical analyses

We provide demographic data and laboratory concentrations using proportions, mean ± standard deviation (SD) or median with 25th and 75th percentile, as applicable, stratified by region. In separate logistic regression models accounting for age, sex, region, ZCTA, and community death rate, we evaluated the association between the selected laboratory covariate and seropositivity for SARS-CoV-2. We modeled all laboratory covariates as continuous variables either imposing a linear relationship with the logit of seropositivity for SARS-CoV-2 or assuming a non-linear relationship and using restricted cubic splines [27]. The best model fit was determined from an analysis of binned residuals [28]. We plotted odds ratios of seropositivity across the span of the laboratory range for a meaningful decrease in the laboratory value as well as in comparison to a fixed reference value. We set the reference value on the basis of the seminal analysis by Lowrie and Lew [17] describing the association of mortality with commonly measured laboratory concentrations in patients on dialysis. For two laboratory concentrations not assessed in this original work, we considered Xia et al. [20] to assign the hemoglobin reference at 10 g/dL and Tentori et al. [29] to assign the parathyroid hormone reference at 300 pg/mL. For serum albumin, we report monthly median (25th, 75th percentile) concentrations from January to July by SARS-CoV-2 seropositivity status in July; we used the using Wilcoxon rank-sum test to compare the distributions. We used multiple imputation by age group to account for missing covariate data. Data analyses were performed using SAS and Stata.

## Results

Of the 28,503 patients in the sample two did not have data on residence zip code and were thus dropped from further analysis; 8% (n = 2292) were seropositive for SARS-CoV-2 total antibody in July 2020. Approximately half of our sampled patients were aged 65 years or above, and the majority (57%) were men (Table 1). There were fewer octogenarians in the South, and

**Table 1. Sampled patient characteristics by region.**

| Patient characteristics | Selected Sample | Northeast | South | Midwest | West |
|---|---|---|---|---|---|
| | N = 28501 | N = 4536 | N = 10937 | N = 3763 | N = 9265 |
| **Age** | | | | | |
| 18–44 | 3,303 (11.6) | 439 (9.7) | 1337 (12.2) | 399 (10.6) | 1128 (12.2) |
| 45–64 | 11539 (40.5) | 1683 (37.1) | 4703 (43.0) | 1417 (37.7) | 3736 (40.3) |
| 65–79 | 10220 (35.9) | 1749 (38.6) | 3831 (35.0) | 1449 (38.5) | 3191 (34.4) |
| ≥80 | 3439 (12.1) | 665 (14.7) | 1066 (9.8) | 498 (13.2) | 1210 (13.1) |
| **Sex** | | | | | |
| M | 16348 (57.4) | 2626 (57.9) | 6194 (56.6) | 2101 (55.8) | 5427 (58.6) |
| F | 12153 (42.6) | 1910 (42.1) | 4743 (43.4) | 1662 (44.2) | 3838 (41.4) |
| Home therapy^ | 1788 (6.3) | 208 (5.5) | 105 (2.3) | 737 (6.7) | 738 (8.0) |
| **Race and Ethnicity** | | | | | |
| Hispanic | 3187 (11.2) | 119 (2.6) | 1641 (15.0) | 83 (2.2) | 1344 (14.5) |
| Non-Hispanic white | 6532 (22.9) | 890 (19.6) | 2868 (26.2) | 1100 (29.2) | 1674 (18.1) |
| Non-Hispanic Black | 4893 (17.2) | 880 (19.4) | 2963 (27.1) | 524 (13.9) | 526 (5.7) |
| Non-Hispanic Other | 2479 (8.7) | 280 (6.2) | 253 (2.3) | 78 (2.1) | 1868 (20.2) |
| Unknown | 11410 (40.0) | 2367 (52.2) | 3212 (29.4) | 1978 (52.6) | 3853 (41.6) |
| **ZCTA Majority Race and Ethnicity[+]** | | | | | |
| Non-Hispanic white | 8733 (30.6) | 1435 (31.6) | 3329 (30.4) | 2271 (60.4) | 1698 (18.3) |
| Non-Hispanic Black | 2585 (9.1) | 564 (12.4) | 1276 (11.7) | 670 (17.8) | 75 (0.8) |
| Hispanic | 4568 (16.0) | 383 (8.4) | 2263 (20.7) | 114 (3.0) | 1808 (19.5) |
| Hispanic and Black | 2878 (10.1) | 734 (16.2) | 1488 (13.6) | 139 (3.7) | 517 (5.6) |
| Other | 9737 (34.2) | 1420 (31.3) | 2581 (23.6) | 569 (15.1) | 5167 (55.8) |
| **Laboratory concentrations** | | | | | |
| Hemoglobin, g/dL | 10.7 (1.4) | 10.5 (1.4) | 10.7 (1.5) | 10.7 (1.5) | 10.8 (1.4) |
| Missing | 33 (0.1) | 4 (0.1) | 12 (0.1) | 3 (0.1) | 14 (0.2) |
| Potassium, mEq/L | 4.7 (0.7) | 4.7 (0.7) | 4.7 (0.7) | 4.6 (0.7) | 4.8 (0.7) |
| Missing | 56 (0.2) | 8 (0.2) | 31 (0.3) | 1 (0.03) | 16 (0.2) |
| Phosphorus, mg/dL | 5.60 (1.8) | 5.48 (1.7) | 5.65 (1.8) | 5.49 (1.7) | 5.6 (1.8) |
| Missing | 46 (0.2) | 10 (0.2) | 0.17 (0.2) | 1 (0.03) | 18 (0.2) |
| Creatinine, mg/dL | 8.61 (3.3) | 8.74 (3.4) | 8.62 (3.4) | 8.03 (3.3) | 8.8 (3.2) |
| Missing | 59 (0.2) | 10 (0.2) | 30 (0.3) | 3 (0.1) | 16 (0.2) |
| Sodium, mEq/L | 138.2 (3.8) | 138.2 (3.8) (3.83) | 138.6 (3.7) (3.70) | 138.4 (3.9) (3.92) | 137.5 (3.8) |
| Missing | 126 (0.4) | 12 (0.3) | 88 (0.8) | 9 (0.2) | 17 (0.2) |
| Albumin, g/dL | 3.7 (0.4) | 3.7 (0.4) | 3.7 (0.4) | 3.7 (0.4) | 3.8 (0.4) |
| Missing | 140 (0.5) | 7 (0.2) | 17 (0.2) | 102 (2.7) | 14 (0.2) |
| PTH[*], pg/mL | 350 (209,554) | 356(209,575) | 351 (211,550) | 322 (194,509) | 361 (217,579) |
| Missing | 4886 (17.1) | 290 (6.4) | 1248 (11.4) | 215 (5.7) | 3133 (33.8) |

Table reports count (percent) or mean (SD) except as noted.

^a participant was defined as on a home modality if he/she had measures of weekly total kt/v *Median (25th, 75th percentile).

[+]ZCTA Majority defined as population in ZCTA ≥ 60% Hispanic, Non-Hispanic Black, or Non Hispanic White; if in remainder ZCTAs Hispanic and Black population exceeded ≥60%, ZCTA defined as 'Hispanic and Black', else as 'Other'. Abbreviations: ZCTA-zip code tabulation area, PTH-parathyroid hormone.

lower mean serum creatinine concentrations were observed in the Midwest. There was substantially higher missingness of PTH concentrations in the West.

## Relations with laboratory values

In adjusted models comparing to a reference value, odds of seropositivity were higher among patients with serum albumin below 4 g/dL, serum creatinine below 12.5 mg/dL, and hemoglobin below 10 g/dL (Fig 1 Panels A-C). Odds of seropositivity for serum albumin 3 and 3.5 g/dL were 2.1 (95% CI 1.9–2.3) and 1.3 (1.2–1.4) respectively, compared with 4 g/dL. Odds of seropositivity for serum creatinine 5 and 8 mg/dL were 1.8 (1.6–2.0) and 1.3 (1.2–1.4) respectively, compared with 12.5 mg/dL.

In adjusted continuous models, serum albumin and creatinine correlated inversely with odds of SARS-CoV-2 seropositivity (Fig 1 Panel D,E). For example, odds of seropositivity were 58% higher in a patient with serum albumin 3.0 g/dL compared to a patient with serum albumin 3.5 g/dL. Odds of seropositivity were 12% higher in patients with serum creatinine 6 mg/dL compared to patients with serum creatinine 7 mg/dL. The relation plateaued above a threshold, such that relative differences in likelihood of seropositivity comparing serum albumin 4.0 g/dL versus 4.5 g/dL and serum creatinine 11 g/dL versus 12 g/dL were minimal. For hemoglobin, there was a suggestion of a bimodal relationship such that odds of seropositivity were highest with hemoglobin 9–10 g/dL (Fig 1 Panel F).

Serum sodium and potassium concentrations, serum phosphate, and PTH correlated inversely with odds of seropositivity (S1 and S2 Figs). Compared to a referent potassium at 5 meq/L, however, while patients with lower serum potassium had higher odds of seropositivity,

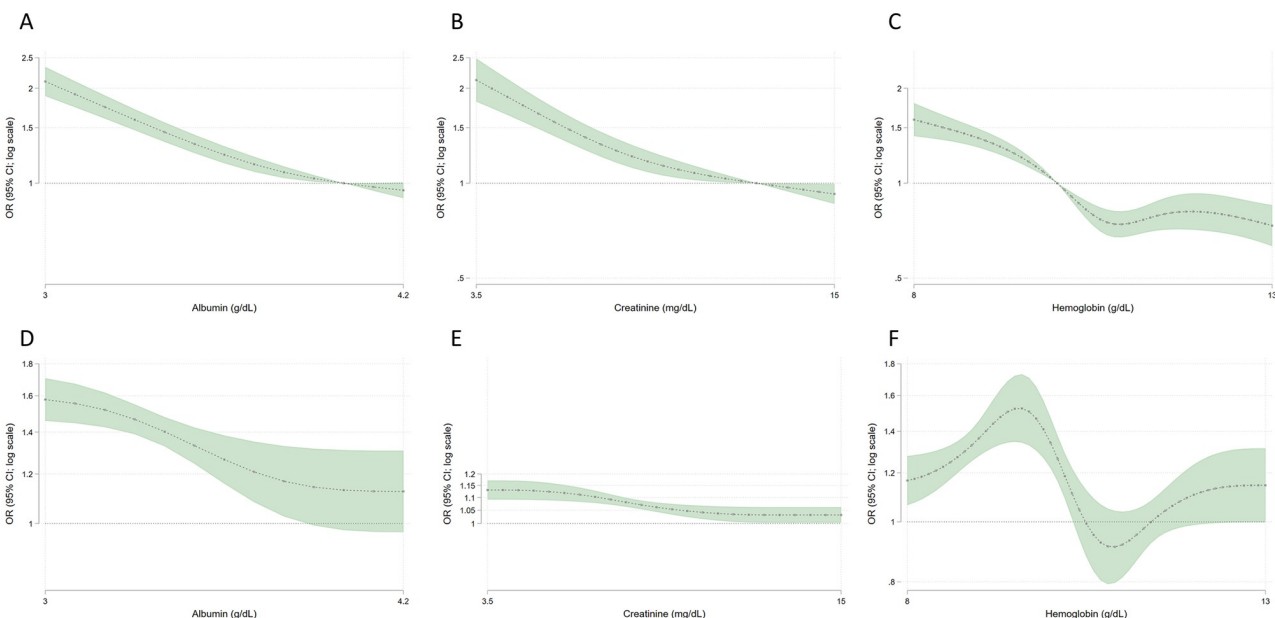

**Fig 1. Relations among three laboratory surrogates of health status and SARS-CoV-2 seropositivity.** Panels A, B, and C show odds of seropositivity compared with reference values of 4 g/dL, 12.5 g/dL, 10 g/dL for albumin, creatinine, and hemoglobin, respectively. For example, in panel A odds of seropositivity for albumin 3 and 3.5 g/dL were 2.1 (95% CI 1.9–2.3) and 1.3 (1.2–1.4) respectively, compared with 4 g/dL. Panels D, E, and F show odds of seropositivity relative to 0.5 g/dL, 1 mg/dL, and 1 g/dL decrease in serum albumin, creatinine, and hemoglobin, respectively. For example, in panel E, when creatinine equals 5 mg/dL, the plot shows the odds ratio (OR) comparing a decrease in creatinine from 5 mg/dL to 4 mg/dL. Models account for differences in SARS-CoV-2 seroprevalence by age, sex, region, county level deaths per 100,000 from SARS-CoV-2, and zip code tabulation area racial or ethnic mix.

these odds were also higher for seropositivity at potassium concentrations above 5.0 meq/L (S1 Fig Panel 1A).

We also evaluated whether age modified the association between albumin and creatinine, and seropositivity. We found that in general older patients had higher risk for SARS-CoV-2 seropositivity than younger patients below the referent values (S3 and S4 Figs).

### Distribution of antecedent serum albumin concentrations

In comparing the distribution of monthly serum albumin concentrations stratified by SARS-CoV-2 seropositive status in July, we found that medians were identical (3.8 g/dL) in seropositive and seronegative groups prior to the emergence of the pandemic in the U.S. (Fig 2). In the seropositive group, albumin concentrations dropped during the first SARS-CoV-2 wave (March-April-May). The nadir occurred in May, with median albumin concentration 3.6g/dL ($25^{th}$, $75^{th}$ percentile: 3.3, 3.9 g/dL) versus 3.8g/dL ($25^{th}$, $75^{th}$ percentile: 3.5,4.0 g/dL) among seropositive versus seronegative patients, respectively (p-value < 0.0001). Consistently, in January 2020, similar proportions of patients with and without seropositivity had albumin < 3.5 g/dL (23 versus 22%, p = 0.21). By May 2020, 39% of patients with versus 22% of patients without SARS-CoV-2 seropositivity had serum albumin < 3.5 g/dL (p < 0.0001).

### Discussion

In this analysis of laboratory correlates of SARS-CoV-2 seropositivity among patients on dialysis, we find that after accounting for community burden of COVID-19, patients with poorer

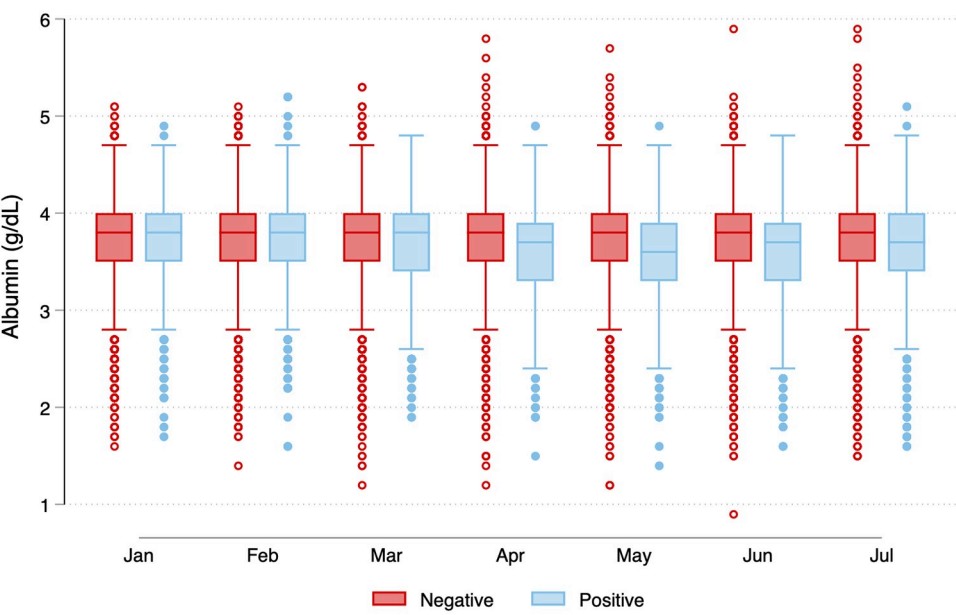

**Fig 2. Serum albumin concentrations among patients with and without SARS-CoV-2 antibodies in July 2020.**
Among patients who did not have SARS-CoV-2 antibodies in July, the serum albumin concentrations remained stable between January and July 2020 (median 3.8 [$25^{th}$, $75^{th}$ percentile, 3.5, 4.0] g/dL). For patients with SARS-CoV-2 seropositivity in July, median serum albumin concentrations were identical to the group without antibodies in January and February. However in the group with seropositivity in July, median albumin concentrations started to drop in March, with the lowest concentrations observed in May (median 3.6 [$25^{th}$, $75^{th}$ percentile 3.3, 3.9] g/dL). In any given month, a patient could contribute more than one observation as multiple laboratory draws were possible. For patients seronegative for SARS-CoV-2 in July 2020, N by month were 22395, 23127, 24035, 24612, 25369, 25994, and 26082 from January to July 2020. For patients seronegative for SARS-CoV-2 in July 2020, N by month was 22395, 23127, 24035, 24612, 25369, 25994, and 26082 from January to July 2020. For patients seropositive for SARS-CoV-2 in July 2020, N by month was 1694, 1772, 1949, 2036, 2131, 2259, and 2284 from January to July 2020. *P-value from Wilcoxon rank-sum test comparing the distribution of monthly concentrations by SARS-CoV-2 seropositivity status.

health status (as reflected by lower serum concentrations of creatinine, albumin, and hemoglobin) have higher likelihood of SARS-CoV-2 antibody. These findings were contrary to our hypothesis, as we had proposed that patients with more robust health status would have a higher likelihood of seropositivity, either due to survivor bias, enhanced ability to mount an antibody response, or lower adherence to social distancing measures. Instead we found that patients with SARS-CoV-2 antibodies also have lower overall health status as measured by laboratory surrogates.

Our findings match those from a recent analysis of risk factors for diagnosis of SARS-CoV-2 as reported to the French national registry of patients on dialysis (REIN) [30]. In this analysis, all patients with consistent symptoms as well as diagnostic rtPCR over a 3-month period were compared to patients without diagnosed SARS-CoV-2 infection. A substantial proportion (~41%) were outpatients with mild or no symptoms. Frail patients—e.g., those requiring assistance to transfer, or those with ischemic heart disease or serum albumin < 3.5 g/dL— experienced higher odds of infection.

Our retrospective analyses suggest one potential explanation for this finding. It has long been known that viral respiratory infections take a metabolic toll, decreasing serum albumin [31–33]. Although SARS-CoV-2 infection is reported to be asymptomatic in up to 40% of patients on dialysis [13], it may nonetheless trigger a significant inflammatory response [34, 35], as evidenced by the drop in serum albumin concentrations among patients with SARS-CoV-2 seropositivity concurrent with the Spring peak of the pandemic in the U.S. In fact, our observed effect may be attenuated because patients who were contemporaneously hospitalized or died from SARS-CoV-2 would have been expected to have the sharpest drop in serum albumin, and would not have been among the patients sampled. This implies that even among the survivors, SARS-CoV-2 has had a significant adverse effect on health status, since patients on dialysis with albumin <3.5 g/dL have a 7-fold higher risk for death, compared with patients with albumin ≥ 4g/dL [11].

Alternatively, in the event of SARS-CoV-2 exposure, contrary to the documented lower likelihood of antibody response after hepatitis B and influenza vaccination [36, 37], patients with poorer health status may be more likely to experience illness associated with an antibody response. Some reports assessing quantitative antibody responses following SARS-CoV-2 infection illustrate that patients without symptoms or with milder disease may be less likely to mount an antibody response, or if a response is present, it may be weaker and of shorter duration [38–40]. In an analysis of 635 serial samples of persons with SARS-CoV-2, 13% of persons who were asymptomatic or remained outpatient did not have evidence of receptor binding domain IgG at four weeks, compared with only 2% of persons hospitalized with SARS-CoV-2 [38]. Furthermore, there was a faster slope of decline in antibody titers among persons with mild or no illness. Thus, it is possible that healthier patients receiving dialysis may have had similar rates of infection, but in the cross-section, due to their milder disease course, less likely to have evidence of antibody.

Other markers associated with higher likelihood of seropisitivity included lower serum sodium concentrations, serum phosphate, and PTH concentrations. A single (baseline) measure of low serum sodium is associated with higher mortality among patients on dialysis [41], and may reflect fluid overload or heart failure—although data on these comorbidities were not available in our analysis. Low serum phosphate and parathyroid hormone concentrations both reflect poorer nutritional status. In their analysis of 622 patients on dialysis followed for over 14 years, Avram et al. found that low PTH concentrations were associated with higher mortality risk, and that a 10-fold increase in PTH was associated with a 27% lower risk for mortality [42]. These findings are confirmed by others among patients on dialysis [42–45], indicating

that extremes of PTH and phosphate concentrations also reflect poorer health status among patients on dialysis.

Our analyses are limited by lack of data on rtPCR-proven diagnosis, symptoms, or hospitalizations. Furthermore, we did not have access to data on dialysis prescription, clinical status, comorbidities, nursing home residence or functional status. The associations assessed here are cross-sectional, and further longitudinal analyses are required to definitively determine whether the purported associations are due to, versus predispose to, SARS-CoV-2 infection. Our analysis also has several strengths, including the use of a central laboratory, a well-validated assay for SARS-CoV-2 seropositivity, retrospective data on serum albumin concentrations, and a large sample size with information to account for other factors associated with seropositivity.

In summary, laboratory concentrations consistently associated with poorer health status and higher risk for mortality were also associated with higher likelihood of SARS-CoV-2 antibodies in persons receiving dialysis. Further insights will be gained by examining quantitative and longitudinal patient samples to assess the risk factors for infection, and the strength and duration of antibody response post-infection.

## Supporting information

**S1 Fig. Relations among serum potassium and sodium and SARS-CoV-2 seropositivity.** Serum potassium (Panel A) and sodium (Panel B) concentrations < 5 meq/L and < 140 meq/L respectively were associated with higher odds of seropositivity. For potassium, there was also a higher odds for seropositivity at potassium concentrations above 5.0 meq/L. (TIF)

**S2 Fig. Relations among serum phosphate and PTH, and SARS-CoV-2 seropositivity.** Panels A and B show odds of seropositivity compared with reference concentrations of 5.5 mmol/L and 300 pg/mL for phosphate and PTH respectively. (TIF)

**S3 Fig. Relations among albumin and SARS-CoV-2 seropositivity by age groups.** In evaluating whether age modified the association between albumin and seropositivity, we found that the overall relationship was similar across age categories. Older patients however had higher odds of SARS-CoV-2 seropositivity than younger patients at serum albumin concentrations below 4 g/dL vs. a concentration at 4 g/dL (p value for interaction = 0.0035). (TIF)

**S4 Fig. Relations among creatinine and SARS-CoV-2 seropositivity by age groups.** For serum creatinine, a similar trend holds true, i.e., that overall patients with lower serum creatinine had higher odds for seropositivity, and that older patients had higher odds of SARS-CoV-2 seropositivity than younger patients at serum creatinine concentration 12.5 mg/dL (p value for interaction = 0.0058). However among the 80 years or above category, a small number of persons had serum creatinine concentration 12.5 mg/dL or higher, thereby obscuring this trend. (TIF)

## Author Contributions

**Conceptualization:** Shuchi Anand, Russell Kerschmann, Paul Beyer, Glenn M. Chertow.

**Data curation:** Maria E. Montez-Rath, Pablo Garcia, Julie Bozeman, Russell Kerschmann.

**Formal analysis:** Maria E. Montez-Rath, Jialin Han, Pablo Garcia.

**Funding acquisition:** Julie Bozeman, Russell Kerschmann, Paul Beyer.

**Investigation:** Shuchi Anand.

**Methodology:** Shuchi Anand, Maria E. Montez-Rath, Pablo Garcia, Julie Bozeman, Russell Kerschmann, Paul Beyer, Julie Parsonnet, Glenn M. Chertow.

**Project administration:** Maria E. Montez-Rath, Jialin Han, Julie Bozeman, Russell Kerschmann, Paul Beyer, Glenn M. Chertow.

**Resources:** Julie Bozeman.

**Software:** Jialin Han.

**Supervision:** Maria E. Montez-Rath, Russell Kerschmann, Paul Beyer, Julie Parsonnet.

**Writing – original draft:** Shuchi Anand, Julie Parsonnet, Glenn M. Chertow.

**Writing – review & editing:** Pablo Garcia, Julie Parsonnet.

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
