## [Decision Letter · Decision Letter 0]

19 Jan 2021

PONE-D-20-36552

Laboratory Correlates of SARS-CoV-2 Seropositivity in a Nationwide Sample of Patients on Dialysis in the U.S.Laboratory Correlates of SARS-CoV-2 Seropositivity in a Nationwide Sample of Patients on Dialysis in the U.S.

PLOS ONE

Dear Dr. Anand,

Thank you for submitting your manuscript to PLOS ONE. After careful consideration, we feel that it has merit but does not fully meet PLOS ONE’s publication criteria as it currently stands. Therefore, we invite you to submit a revised version of the manuscript that addresses the points raised during the review process.

We look forward to receiving your revised manuscript.

Kind regards,

Sherief Ghozy, M.D., Ph.D. candidate

Academic Editor

PLOS ONE

Journal Requirements:

2. Thank you for including your ethics statement: "Stanford University IRB approved the study."   

3. In your ethics statement in the manuscript and in the online submission form, please ensure that you have discussed whether all data/samples were fully anonymized before you accessed them and/or whether the IRB or ethics committee waived the requirement for informed consent. If patients provided informed written consent to have data/samples from their medical records used in research, please include this information.

4. In the ethics statement in the manuscript and in the online submission form, please provide additional information about the patient records/samples used in your retrospective study, including: a) the date range (month and year) during which patients' medical records/samples were accessed; b) the date range (month and year) during which patients whose medical records/samples were selected for this study sought treatment; and c) the source of the medical records/samples analyzed in this work (e.g. hospital, institution or medical center name).

5. During your revisions, please note that a simple title correction is required: Please ensure the title is entered only once in online submission form, so that it reads "Laboratory Correlates of SARS-CoV-2 Seropositivity in a Nationwide Sample of Patients on Dialysis in the U.S."

6. Thank you for stating the following in the Competing Interests section:

We note that one or more of the authors are employed by a commercial company: Ascend Clinical Laboratory.

6.1. Please provide an amended Funding Statement declaring this commercial affiliation, as well as a statement regarding the Role of Funders in your study. If the funding organization did not play a role in the study design, data collection and analysis, decision to publish, or preparation of the manuscript and only provided financial support in the form of authors' salaries and/or research materials, please review your statements relating to the author contributions, and ensure you have specifically and accurately indicated the role(s) that these authors had in your study. You can update author roles in the Author Contributions section of the online submission form.

6.2. Please also provide an updated Competing Interests Statement declaring this commercial affiliation along with any other relevant declarations relating to employment, consultancy, patents, products in development, or marketed products, etc.  

7. We note that you have indicated that data from this study are available upon request. PLOS only allows data to be available upon request if there are legal or ethical restrictions on sharing data publicly. For information on unacceptable data access restrictions, please see http://journals.plos.org/plosone/s/data-availability#loc-unacceptable-data-access-restrictions.

Reviewers' comments:

Reviewer's Responses to Questions

**Comments to the Author**

1. Is the manuscript technically sound, and do the data support the conclusions?

Reviewer #1: Yes

Reviewer #2: No

Reviewer #3: Yes

2. Has the statistical analysis been performed appropriately and rigorously? 

Reviewer #1: Yes

Reviewer #2: Yes

Reviewer #3: Yes

3. Have the authors made all data underlying the findings in their manuscript fully available?

Reviewer #1: Yes

Reviewer #2: Yes

Reviewer #3: Yes

4. Is the manuscript presented in an intelligible fashion and written in standard English?

Reviewer #1: Yes

Reviewer #2: Yes

Reviewer #3: Yes

5. Review Comments to the Author

Reviewer #1: I read with great interest the paper entitled "Laboratory Correlates of SARS-CoV-2 Seropositivity in a Nationwide Sample of Patients on Dialysis in the U.S".

This cross-sectional study aimed to assess the association between SARS-CoV-2 antibodies and seven laboratory (inflammatory and nutritional) parameters in 28,503 dialysis patients. Authors hypothesized that patients with better nutritional laboratory parameters would be able to mount their antibodies, however they found more positive serology in patients with parameters associated with poorer health status.

The research question is clear and limitations of the study were clearly stated at the end.

The strength of this study resides in the large sample and the analysis of serum albumin over several months before the SARS-CoV-2 seropositivity. Obviously, due to the cross-sectional design of the study, one cannot draw causal inference.

I have some comments that could help the authors improve their manuscript, if they can address them in the methodology, discussion and/or limitations.

Major comments:

1-Why did the authors consider only the serum albumin as highly affected by inflammation? Hemoglobin may drop in acute infections and the speed of this drop can vary between an individual and another. It would have been interesting to look at hemoglobin with the same strategy followed for serum albumin between January and July. Same thoughts for potassium and phosphate that can be affected by an acute infection and drop within days if the patient is sick and not eating. How do authors interpret the association between seropositivity and low PTH (this was not addressed in the discussion)? Why did authors choose creatinine and not BUN? Why did authors assume that sodium can be assessed without any clinical data (fluid overload for instance)?

Minor comments: I couldn't find within the manuscript the percentage of hemodialysis versus peritoneal dialysis patients.

Reviewer #2: The authors focused on the relationship laboratory data and COVID-19 seropositivity. This study is a relatively large-scale observational study, and the significance of the obtained knowledge is great. However, I think that the results are not well discussed in manuscript.

My comments are as follows.

Major

1. Before the pandemic, most of the patients were mild status, whereas after the pandemic, there should be many severe cases. Patients with high antibody titers are more likely to have been hospitalized due to severe illness, and their serum albumin is likely to have decreased during the course of treatment for the infection.

2. In Figure 2, the number of patients per graph should be described.

3. There is no difference in serum albumin value before the pandemic between the antibody positive and negative cases, and the serum albumin is low after the pandemic. Since this paper is an observational study, it should be avoided to mention that low albumin is associated with mortality.

4. Elevated antibody titers are the result of infection, and health before infection is important. Comparing positive and negative antibody titers does not directly compare the risk of infection.

Minor

1. The author should describe what kind of statistical analysis was done for the content stated in the last sentence of Result section.

2. There are BCG method, BCP method and BCP improvement method as albumin level measurement methods. It should be shown whether all facilities use the same testing method.

Reviewer #3: The study is interesting, and it clearly states the results found in it. However, I have some remarks and recommendations for it.

1. It should state the criteria of the authors to select the cut-off points to make the comparison among the different variables, e.g. creatine 12,5 vs 8 vs 5.

2. Having a wide range of the participants' age would make it possible to perform a sub-analysis with different sublayers to evaluate the behavior of the chosen variables according to the patients' age.

3. It is shown to detail the results of the albumin in May and the difference between the two studied groups. But it is unknown if this variation happens also with the other variables, or it is exclusive to the albumin.

4. The hypotheses to find the variation of the albumin are clearly and meticulously state in the article. Nevertheless, there is no enough information to support the hypothesis for the variation of the other variables.

5. In graph B from Figure 1, the maximum value of the creatinine is missing.

6. In Figure 2, there is a typo mistake, the abbreviation of February is Feb instead of Fev.

7. Evaluating calcium involvement as a variable in the analysis. There several studies that relate it with a mineral metabolic bone disease, and this one with an alteration of the immune system.

8. The Figures that show the behavior of the Sodium and PTH reveal that having high values of them reduces the probability of acquiring antibodies. I think it should be some comments on the results and hypothesis about it.

I really enjoyed reading this article, congratulations to all the authors for such an interesting paper.

6. PLOS authors have the option to publish the peer review history of their article (what does this mean?). If published, this will include your full peer review and any attached files.

Reviewer #1: **Yes: **Mabel Aoun

Reviewer #2: No

Reviewer #3: No

---

## [Author Response · Author response to Decision Letter 0]

16 Mar 2021

PONE-D-20-36552

Laboratory Correlates of SARS-CoV-2 Seropositivity in a Nationwide Sample of Patients on Dialysis in the U.S.Laboratory Correlates of SARS-CoV-2 Seropositivity in a Nationwide Sample of Patients on Dialysis in the U.S.

Response to Review

Journal Requirements:

Response: These have been addressed, and the paper reformatted accordingly. 

2. Thank you for including your ethics statement: "Stanford University IRB approved the study." 

Response: The statement has been revised to state that the Stanford University Institutional Review Board 61 (Registration 4947) approved this work under study protocol #56901 (Lines 94-99). 

3. In your ethics statement in the manuscript and in the online submission form, please ensure that you have discussed whether all data/samples were fully anonymized before you accessed them and/or whether the IRB or ethics committee waived the requirement for informed consent. If patients provided informed written consent to have data/samples from their medical records used in research, please include this information.

Response: Thank you for the careful review. We added below to our Revised Materials and Methods: 

The data were fully anonymized before Stanford University researchers analyzed them; all sample analyses were performed as part of routine clinical care or using plasma that would have otherwise been discarded (for the SARS-CoV-2 antibody testing). The Stanford University IRB waived requirement for informed consent (Lines 94-99).

4. In the ethics statement in the manuscript and in the online submission form, please provide additional information about the patient records/samples used in your retrospective study, including: a) the date range (month and year) during which patients' medical records/samples were accessed; b) the date range (month and year) during which patients whose medical records/samples were selected for this study sought treatment; and c) the source of the medical records/samples analyzed in this work (e.g. hospital, institution or medical center name).

Response: In the requested place we have now updated these data to indicate: All data were anonymized prior to analysis Patients’ electronic health data on demographics and comorbidities as available to Ascend Clinical were accessed for the month of the seroprevalence analysis (July 2020). Patients’ laboratory data—performed as part of routine clinical care—were accessed dating back 6 months prior to seroprevalence testing (January 2020). The source of data were Ascend Clinical laboratory and electronic medical record data (See Revised Methods Lines 107-112). 

5. During your revisions, please note that a simple title correction is required: Please ensure the title is entered only once in online submission form, so that it reads "Laboratory Correlates of SARS-CoV-2 Seropositivity in a Nationwide Sample of Patients on Dialysis in the U.S."

Response: This correction has been made. 

6. Thank you for stating the following in the Competing Interests section:

We note that one or more of the authors are employed by a commercial company: Ascend Clinical Laboratory.

Response: This is correct; thank you again for your careful review. We have updated the Competing interests section to address this (see 6.1). 

6.1. Please provide an amended Funding Statement declaring this commercial affiliation, as well as a statement regarding the Role of Funders in your study. If the funding organization did not play a role in the study design, data collection and analysis, decision to publish, or preparation of the manuscript and only provided financial support in the form of authors' salaries and/or research materials, please review your statements relating to the author contributions, and ensure you have specifically and accurately indicated the role(s) that these authors had in your study. You can update author roles in the Author Contributions section of the online submission form.

Response: We updated the funding statement to state clearly that:

Ascend Clinical funded the remainder plasma testing performed for seroprevalence analysis. In addition, authors JB, RK, and PB are employed by Ascend Clinical. JB assisted with data preparation including anonymized electronic health data preparation, RK selected the laboratory assays and supervised sample analysis, and PB co-conceived the study with GMC. 

6.2. Please also provide an updated Competing Interests Statement declaring this commercial affiliation along with any other relevant declarations relating to employment, consultancy, patents, products in development, or marketed products, etc. 

Response: We have updated the Competing interests statement to the following: 

JB, RK and PB are employed by Ascend Clinical Laboratories. GMC is on the Board of Satellite Healthcare, a not-for-profit dialysis organization. This does not alter our adherence to PLoS ONE policies on sharing data and materials. 

Response: We certify that all competing interests have been declared, and we thank PLoS One Editors for helping us to clearly state all competing interests. 

7. We note that you have indicated that data from this study are available upon request. PLOS only allows data to be available upon request if there are legal or ethical restrictions on sharing data publicly. For information on unacceptable data access restrictions, please see http://journals.plos.org/plosone/s/data-availability#loc-unacceptable-data-access-restrictions.

Response: We do not have an objection to making the data publically available, and have prepared a data set for publication on our website, at a timeline suggested by the Editors: https://covidkidney.stanford.edu/

Response: We have prepared a dataset to be publically available at our website: https://covidkidney.stanford.edu/ . We will await instructions from the Editorial team regarding the timing of its publication. 

Reviewers' comments:

Reviewer's Responses to Questions

Comments to the Author

1. Is the manuscript technically sound, and do the data support the conclusions?

Reviewer #1: Yes

Reviewer #2: No

Reviewer #3: Yes

2. Has the statistical analysis been performed appropriately and rigorously? 

Reviewer #1: Yes

Reviewer #2: Yes

Reviewer #3: Yes

3. Have the authors made all data underlying the findings in their manuscript fully available?

Reviewer #1: Yes

Reviewer #2: Yes

Reviewer #3: Yes

4. Is the manuscript presented in an intelligible fashion and written in standard English?

Reviewer #1: Yes

Reviewer #2: Yes

Reviewer #3: Yes

5. Review Comments to the Author

 Reviewer #1: I read with great interest the paper entitled "Laboratory Correlates of SARS-CoV-2 Seropositivity in a Nationwide Sample of Patients on Dialysis in the U.S".

This cross-sectional study aimed to assess the association between SARS-CoV-2 antibodies and seven laboratory (inflammatory and nutritional) parameters in 28,503 dialysis patients. Authors hypothesized that patients with better nutritional laboratory parameters would be able to mount their antibodies, however they found more positive serology in patients with parameters associated with poorer health status.

The research question is clear and limitations of the study were clearly stated at the end.

The strength of this study resides in the large sample and the analysis of serum albumin over several months before the SARS-CoV-2 seropositivity. Obviously, due to the cross-sectional design of the study, one cannot draw causal inference.

I have some comments that could help the authors improve their manuscript, if they can address them in the methodology, discussion and/or limitations.

Major comments:

1-Why did the authors consider only the serum albumin as highly affected by inflammation? Hemoglobin may drop in acute infections and the speed of this drop can vary between an individual and another. It would have been interesting to look at hemoglobin with the same strategy followed for serum albumin between January and July. Same thoughts for potassium and phosphate that can be affected by an acute infection and drop within days if the patient is sick and not eating. 

Response: We thank the Reviewer for his or her careful and overall positive review. We agree with the Reviewer that in this cross-sectional analysis, it is hard to disentangle factors that could be responsible for the findings. As astutely implied by the Reviewer, it is possible that patients’ with SARS-CoV-2 infection and resultant seropositivity had a decline in concentrations of albumin, hemoglobin, potassium, and/or phosphate and that concentrations of albumin, hemoglobin, potassium, and/or phosphate played no role in developing COVID-19 or mounting an antibody response to SARS-CoV-2. 

In our discussion (paragraphs 3 and 4, lines 237-259) we posit that either scenario could exist to explain our observed associations. Our limited retrospective analyses using a selected marker (i.e., serum albumin) seem to support the first (i.e., a decline in serum albumin occurred around the time of illness). We selected serum albumin as the primary marker of health status in our analyses due to: 1) its likely variation in response to infection and inflammatory response, and 2) its strong and consistently observed inverse association with mortality among patients on dialysis. The other markers—e.g., phosphate or parathyroid hormone—have been observed to have a ‘U’ shaped relationship with mortality among patients on dialysis (Block et al. JASN 2004) . We added this explanation and appropriate references to our methods to strengthen our rationale for selecting serum albumin (see Revised Materials and Methods line 130-132). 

Rather than perform further retrospective analyses to explain this cross-sectional association, we propose (and are carrying out) a prospective study to evaluate whether SARS-CoV-2 illness has significant effects on the health status of patients on dialysis. 

2. How do authors interpret the association between seropositivity and low PTH (this was not addressed in the discussion)? Why did authors choose creatinine and not BUN? Why did authors assume that sodium can be assessed without any clinical data (fluid overload for instance)?

Response: Many years ago, Avram and others (AJKD 1996 and 2001) highlighted the association between parathyroid hormone concentrations and nutritional status in patients receiving dialysis. Although PTH can be influenced by many factors, including serum concentrations of calcium and phosphate and the provision of calcitriol or active vitamin D analogs and calcimimetics, nutritional status continues to play a role. We added this explanation and appropriate references to our Revised Discussion lines 264-273. Lower serum creatinine concentrations are consistently associated with mortality in the dialysis population, most likely related to muscle wasting in the setting of little or no residual kidney function. We elected to include serum creatinine as a marker of muscle mass and quality, not kidney function (as would be the case in a general population cohort). Hoping to limit the inquiry to just a handful of laboratory tests (for a more parsimonious presentation), we did not explore associations with BUN. In patients receiving dialysis, the urea nitrogen concentration is confounded by dietary protein intake, the type and integrity of vascular access, and other factors.

On the Reviewer’s last comment on serum sodium, we agree that interpretation of our analyses in general requires caution as we do not have detailed clinical data. Several observational studies have however demonstrated higher mortality among patients on dialysis with ‘baseline’ (single measure) low serum sodium (Sun et al. Scientific Reports 2017). We added a systematic review of these studies as a reference to our Revised Discussion lines 264-273, and have also added the limitation of clinical data to our Revised Discussion line 275 . 

3. Minor comments: I couldn't find within the manuscript the percentage of hemodialysis versus peritoneal dialysis patients.

Response: We did not have direct information on modality, but defined a participant as being on home modality if he/she had measures of weekly total Kt/v—as would be expected for patients on peritoneal dialysis. We have added these data to Table 1. We thank the Reviewer for his or her close reading. 

Reviewer #2: The authors focused on the relationship laboratory data and COVID-19 seropositivity. This study is a relatively large-scale observational study, and the significance of the obtained knowledge is great. However, I think that the results are not well discussed in manuscript.

My comments are as follows.

Major

1. Before the pandemic, most of the patients were mild status, whereas after the pandemic, there should be many severe cases. Patients with high antibody titers are more likely to have been hospitalized due to severe illness, and their serum albumin is likely to have decreased during the course of treatment for the infection.

Response: We agree with the Reviewer that this is one possible explanation for our observed association between serum albumin and SARS-CoV-2 seropositivity. See our text further discussing this potential implication (lines 241-263; 276-278). 

2. In Figure 2, the number of patients per graph should be described.

Response: We thank the Reviewer for his or her close review. Since the figure already had significant data, we added these data to the Figure legend. 

3. There is no difference in serum albumin value before the pandemic between the antibody positive and negative cases, and the serum albumin is low after the pandemic. Since this paper is an observational study, it should be avoided to mention that low albumin is associated with mortality.

Response: We agree and do not claim any inference between albumin and mortality on the basis of our current analyses. However we have cited in Materials and Methods (lines 130-131) prior studies demonstrating this association, to explain our rationale for selecting serum albumin as a chief marker of interest. 

4. Elevated antibody titers are the result of infection, and health before infection is important. Comparing positive and negative antibody titers does not directly compare the risk of infection.

Response: We agree, please see our response to your comment 1 and Reviewer 1’s comment 1. 

Minor

1. The author should describe what kind of statistical analysis was done for the content stated in the last sentence of Result section.

Response: We used the using Wilcoxon rank-sum test to compare the distributions, and this is stated in the methods line149-150. 

2. There are BCG method, BCP method and BCP improvement method as albumin level measurement methods. It should be shown whether all facilities use the same testing method.

Response: The tests were all done at a central laboratory, which used the Bromcresol Green (BCG) methods for albumin testing. We have updated these details in our Materials and Methods section. We again thank the Reviewer for his or her close review of our work. 

Reviewer #3: The study is interesting, and it clearly states the results found in it. However, I have some remarks and recommendations for it.

1. It should state the criteria of the authors to select the cut-off points to make the comparison among the different variables, e.g. creatine 12,5 vs 8 vs 5.

Response: As a reference for the cut points for several laboratory values, we used the seminal analysis published by Lowrie and Lew entitled ‘Death risk in hemodialysis patients: the predictive value of commonly measured variables and an evaluation of death rate differences between facilities’ and published in the American Journal of Kidney Disease in 1990 which first established the relationships between several laboratory markers (including albumin, creatinine, potassium, and phosphate) and risk for death among patients on dialysis. In the Revised Methods line 144-145 we have made clear that we used cut points on the basis of this reference for four laboratories. References for the remainder laboratories are also stated in the Revised Methods line146-147. 

2. Having a wide range of the participants' age would make it possible to perform a sub-analysis with different sublayers to evaluate the behavior of the chosen variables according to the patients' age.

Response: We believe here that the Reviewer is suggesting testing for whether age modifies the association between albumin and seroprevalence for example. We appreciate this suggestion, which is an intriguing perspective on the data. Since our analysis included multiple imputation for missing laboratory correlates, testing for the interaction required us to update the entire analysis. We tested for interaction with the three laboratory correlates featured in our main analysis: albumin, hemoglobin, and creatinine. There was no significant interaction between hemoglobin and age in the association with seroprevalence (p=0.9711). 

Creatinine Albumin Hemoglobin 

We found that for serum albumin, the overall relationship is similar across age categories. Older patients, however, had higher odds of SARS-CoV-2 seroprevalence than younger patients at serum albumin concentrations below 4 g/dL vs a concentration at 4 g/dL (see figure below; p value for interaction = 0.0035). 

For serum creatinine, a similar trend holds true until the age 80 years or above category (see figure below, p value for interaction = 0.0058). The relatively lower muscle mass in patients 80 years or older likely means that the referent serum creatinine level should in fact be lower than 12.5 mg/dL; in fact very few patients over age 80 years old (n=15) had serum creatinine ≥ 12.5 mg/dL. Put in this context, except at the extremes of age groups, the results below continue to support our current conclusion that lower serum creatinine levels are indicative of lower muscle mass and poorer health status in patients on dialysis, and are associated with higher likelihood of SARS-CoV-2 seropositivity. 

We can include these data, figures, and their interpretation in the Supplemental material at the Reviewer or Editor’s discretion. We again thank the Reviewer for his or her careful review. 

3. It is shown to detail the results of the albumin in May and the difference between the two studied groups. But it is unknown if this variation happens also with the other variables, or it is exclusive to the albumin.

Response: Your comment echoes Reviewer 1’s comment #1. As explained in our response to this comment, we relied on serum albumin as our primary marker of interest due to its consistently observed inverse relation with mortality among patients on dialysis. 

4. The hypotheses to find the variation of the albumin are clearly and meticulously state in the article. Nevertheless, there is no enough information to support the hypothesis for the variation of the other variables.

Response: We hope that by adding additional explanations to the observed association for PTH and sodium (see Revised Discussion lines 264-273), we have enriched our discussion as suggested by the Reviewers. 

5. In graph B from Figure 1, the maximum value of the creatinine is missing.

Response: The maximum value is 15, and we have fixed this error. Thank you for your close review of our work. 

6. In Figure 2, there is a typo mistake, the abbreviation of February is Feb instead of Fev.

Response: Again thank you for this close review. We fixed this typo. 

7. Evaluating calcium involvement as a variable in the analysis. There several studies that relate it with a mineral metabolic bone disease, and this one with an alteration of the immune system.

Response: The Reviewer is quite correct that serum calcium is a critical component of the mineral-bone axis in patients on dialysis, which in turn could influence the activity of the immune system. However in this cross sectional analysis we wanted to limit the total number of associations tested, and a priori selected laboratories we hypothesized would have an association with SARS-CoV-2 seropositivity. We had selected two of the three commonly measured mineral bone disease markers (phosphate and PTH), we had not selected a third (calcium). For parsimony, we limited the number of laboratory studies tested to seven. If the Editors feel strongly, we could test calcium, but would prefer to keep the analysis as is.

8. The Figures that show the behavior of the Sodium and PTH reveal that having high values of them reduces the probability of acquiring antibodies. I think it should be some comments on the results and hypothesis about it.

Response: Your comment here is very similar Reviewer 1’s comment 1. We have revised our discussion to include a potential explanation for the observed association (Revised Discussion lines 264-273). 

I really enjoyed reading this article, congratulations to all the authors for such an interesting paper.

We thank the Reviewer for his or her time, and for this encouraging comment. 

6. PLOS authors have the option to publish the peer review history of their article (what does this mean?). If published, this will include your full peer review and any attached files.

Do you want your identity to be public for this peer review? For information about this choice, including consent withdrawal, please see our Privacy Policy.

Reviewer #1: Yes: Mabel Aoun

Reviewer #2: No

Reviewer #3: No

---

## [Editor Report · Decision Letter 1]

19 Mar 2021

Laboratory Correlates of SARS-CoV-2 Seropositivity in a Nationwide Sample of Patients on Dialysis in the U.S.

PONE-D-20-36552R1

Dear Dr. Anand,

We’re pleased to inform you that your manuscript has been judged scientifically suitable for publication and will be formally accepted for publication once it meets all outstanding technical requirements.

Kind regards,

Sherief Ghozy, M.D.

Academic Editor

PLOS ONE

---

## [Editor Report · Acceptance letter]

6 Apr 2021

PONE-D-20-36552R1 

Laboratory correlates of SARS-CoV-2 seropositivity in a nationwide sample of patients on dialysis in the U.S. 

Dear Dr. Anand:

I'm pleased to inform you that your manuscript has been deemed suitable for publication in PLOS ONE. Congratulations! Your manuscript is now with our production department. 

Kind regards, 

on behalf of

Dr. Sherief Ghozy 

Academic Editor

PLOS ONE